# Nitrate Addition Increases the Activity of Microbial Nitrogen Removal in Freshwater Sediment

**DOI:** 10.3390/microorganisms10071429

**Published:** 2022-07-15

**Authors:** Min Cai, Yiguo Hong, Jiapeng Wu, Selina Sterup Moore, Teofilo Vamerali, Fei Ye, Yu Wang

**Affiliations:** 1Institute of Environmental Research at Greater Bay Area, Key Laboratory for Water Quality and Conservation of the Pearl River Delta, Ministry of Education, Guangzhou University, Guangzhou 510006, China; 15883918123@163.com (M.C.); yghong@gzhu.edu.cn (Y.H.); wujiapeng@gzhu.edu.cn (J.W.); 2Department of Agronomy, Food, Natural Resources, Animals and Environment (DAFNAE), University of Padova, 35122 Padova, Italy; selina.moore@unipd.it (S.S.M.); teofilo.vamerali@unipd.it (T.V.)

**Keywords:** external impact, regulation experiments, nitrate, dissolved oxygen, nitrogen removal, nitrous oxide

## Abstract

Denitrification and anammox occur widely in aquatic ecosystems serving vital roles in nitrogen pollution removal. However, small waterbodies are sensitive to external influences; stormwater runoff carrying nutrients and oxygen, flows into waterbodies resulting in a disruption of geochemical and microbial processes. Nonetheless, little is known about how these short-term external inputs affect the microbial processes of nitrogen removal in small waterbodies. To investigate the effects of NO_3_^−^, NH_4_^+^, dissolved oxygen (DO) and organic C on microbial nitrogen removal in pond sediments, regulation experiments have been conducted using slurry incubation experiments and ^15^N tracer techniques in this study. It was demonstrated the addition of NO_3_^−^ (50 to 800 μmol L^−1^) significantly promoted denitrification rates, as expected by Michaelis-Menten kinetics. Ponds with higher NO_3_^−^ concentrations in the overlying water responded more greatly to NO_3_^−^ additions. Moreover, N_2_O production was also promoted by such an addition of NO_3_^−^. Denitrification was significantly inhibited by the elevation of DO concentration from 0 to 2 mg L^−1^, after which no significant increase in inhibition was observed. Denitrification rates increased when organic C was introduced. Due to the abundant NH_4_^+^ in pond sediments, the addition demonstrated little influence on nitrogen removal. Moreover, anammox rates showed no significant changes to any amendment.

## 1. Introduction

Freshwater sediments, especially those of small waterbodies, are sensitive to flow confluence either from human discharge or stormwater runoff [1,2]. During rainfall, runoff accumulate pollutants from farmland, roads and roofs, and deposits them in receiving waterbodies [3]. Many contaminants in runoff, such as soil particles with heavy metals, can deteriorate water quality and lead to a loss of biodiversity [4,5]. Nutrients are commonly observed in runoff, including organic and inorganic compounds [6]. Notably, NO_3_^−^ of high mobility and NH_4_^+^ attached to soil particles are common soluble species and as such are more likely to be brought into small waterbodies [7]. They can have great impact on waterbodies because they are readily absorbed by simple organisms [8], leading to eutrophication and a reduction in biodiversity [9]. Furthermore, organic matter in runoff will consume dissolved oxygen in waters, resulting in negative hypoxia conditions [10], but the higher levels of dissolved oxygen found in runoff due to greater air contact during the flow, can temporarily alleviate states of hypoxia in receiving waterbodies [11]. The nutrients and oxygen carried by the runoff may overwhelmingly alter the physicochemical properties and microbial processes of waterbodies of smaller size [12,13,14].

The nitrogen cycle is recognized as one of the most important biogeochemical processes on earth [15]. Environmental factors such as redox conditions and substrate availability stand to regulate the operation and intensity of these pathways, but these are easily altered and thus highly fluctuating in aquatic systems [16]. Denitrification and anammox are important processes transforming inorganic nitrogen to dinitrogen gas [17] and are vital for the removal of nitrogen pollution in aquatic ecosystems [18,19]. Among the components that are easily carried by runoff into the waterbodies, NO_3_^−^ and NH_4_^+^ provide substrates for denitrification and anammox processes, which can directly affect the microbial nitrogen removal [20,21]. The responses of denitrification and anammox processes to organic C addition are significantly different: being a heterotrophic process, denitrification can use organic C as an electron donor and is thus promoted by the addition of organic C [22]; while anammox, an autotrophic process, is instead reduced by the presence of organic C due to competition with denitrification [23]. The confluence of runoff into waters may bring in dissolved oxygen which can indirectly affect the anaerobic processes of denitrification and annamox by influencing enzyme activities and metabolism [24,25]. Although several studies have correlated environmental factors to microbial nitrogen removal processes in natural habitats [21,26,27], the means by which these factors regulate the intensity of denitrification and anammox activities in small waterbodies remain unknown.

What are the short-term effects of runoff into waterbodies on the microbial nitrogen removal processes? In this context, slurry incubation experiments combined with ^15^N tracer techniques were used in this study to investigate the influence of NO_3_^−^, NH_4_^+^, DO and TOC on nitrogen removal rates in pond sediments. The objective of this study was to (***i***) identify factors influence the denitrification process and to (***ii***) quantify their effects on the sediments of small waterbodies.

## 2. Materials and Methods

### 2.1. Sample Collection and Physicochemical Analysis

Samples were collected from three ponds (pond #1, #2 and #3) located in Chongqing, southwest China (29°56′56″–29°57′43″ N, 106°37′12″–106°38′13″ E) in July 2021. Precipitation in July was 189 mm which was relatively high throughout the year with annual mean of 109 mm in 2021 and resulted in runoff confluence into ponds regularly in this season. At each pond, three parallel surface sediments (0–10 cm) and overlying water were sampled. Field samples were stored in sterile plastic bags and transported to the laboratory in a cooler box (4 °C) for subsequent analysis. One subsample was stored in 4 °C refrigerator for microbial nitrogen removal regulatory experiments, and the second subsample was used for physicochemical analyses. All samples were analyzed in triplicate, and the values were averaged to represent site conditions.

NH_4_^+^, NO_2_^−^, NO_3_^−^ in overlying water were determined via a spectrophotometric detection assay [28]. pH in overlying water was determined with a Mettler Toledo Ph analyzer (S220, Switzerland). Sediment NH_4_^+^, NO_2_^−^, NO_3_^−^ were extracted from 5 g of fresh sediment/soil with 25 mL of 2 M KCl (1:5 *w*/*v*). The supernatant was filtered through a 0.22 μm membrane filter and the compounds were determined via a spectrophotometric detection assay. Moisture content was measured by oven-drying at 105 °C until a constant weight was achieved. The pH was determined in 1:2.5 sediment/water (*w*/*v*) suspensions after shaking and centrifugation, with a Mettler Toledo pH analyzer (S220, Switzerland). Organic matter (OM) was measured as loss on ignition at 550 °C (LOI 550) using a Muffle furnace. The TN and TP were determined with the potassium persulfate oxidation-ultraviolet spectrometry method [29], using a UV spectrophotometer (UVmini-1240, Shimadzu, Japan).

### 2.2. DNA Extraction, Sequencing, and qPCR Analysis

DNA was extracted from sediment samples (approximately 0.5 g) using the FastDNA SPIN Kit for Soil (MP Biomedicals, Irvine, CA, USA) following the manufacturer’s protocol. The concentration of extracted DNA was measured with a NanoDrop Lite (Thermo Fisher Scientific, Wilmington, DE, USA), and the DNA quality determined by means of 1% (*w*/*v*) agarose gel electrophoresis.

The *nirS* gene amplified by PCR using primers cd3aF and R3cd was used to study the denitrification bacterial community [30]. Primers A438f and A684r was used to amplify Anammox-specific 16S rRNA genes [31]. More details about the conditions of PCR amplification are presented in Appendix A. Before high-throughput sequencing, the PCR products were purified using the MiniBEST Agarose Gel DNA Extraction Kit (TaKaRa Bio, Japan). Subsequently, purified amplicons were pooled in equimolar and paired-end (PE) sequenced (2 × 300) on an Illumina MiSeq PE300 platform. Raw sequences were merged and quality filtered in Quantitative Insights in Microbial Ecology (QIIME) [32] and Mothur [33]. OTUs with identity thresholds (93% for *nirS* and 97% for anammox 16S rRNA) were defined by Usearch (v. 7.0 http://drive5.com/uparse/, accessed date: 10 June 2022). Rare OTUs with less than 0.01% of the total sequences were excluded. To avoid biases arising from sequencing depth and to make samples comparable, sequences were rarefied to a uniform sequencing depth based on the sample with the lowest sequences. The raw sequences of *nirS* and Anammox-specific 16S rRNA genes used in this study were deposited in the Sequence Read Archive (SRA, https://submit.ncbi.nlm.nih.gov/subs/sra/, accessed date: 10 June 2022) of NCBI under the accession numbers PRJNA844121.

The abundance *nir* gene (*nirS* and *nirK*), *nosZ* gene (*nosZ* I and *nosZ* II) and anammox-specific 16S rRNA genes in sediments were quantified by a LightCycler^®^ R480 II Real-Time PCR (Roche, Basel, Switzerland). Each sample was analyzed in triplicates. The standard curves used for calculation were achieved with plasmid DNA with known concentrations and copy numbers. Results of qPCR with high amplification efficiency (90–110%) and correlation coefficient values of the standard curve (r^2^ > 0.97) were included in the analysis. The specificity of PCR amplifications was defined by melting curve analysis and gel electrophoresis. Primers, reaction systems and procedures are shown in Appendix A.

### 2.3. Experimental Set Up

Four parallel incubations were performed to investigate the effects of NO_3_^−^, NH_4_^+^, dissolved oxygen (DO) and organic C (as glucose) concentrations on the rates of denitrification, anammox and N_2_O production in pond sediments. In the nitrogen addition experiments, five concentration gradients of ^15^NO_3_^−^ (50, 100, 200, 400 and 800 μmol L^−1^) and five concentration gradients of NH_4_^+^ (0, 20, 40, 80 and 120 μmol L^−1^) were set up in the incubations. The regulation of DO was achieved through replacement, by injecting oxygen-rich water (12 mg L^−1^) to replace the supernatant to reach different DO gradients (0, 1, 2, 4, 6 mg L^−1^). The organic C experiments were set up at 0, 100, 200, 400 and 600 μmol glucose L^−1^, respectively. Four sets of parallel regulation experiments were conducted on three different ponds (pond #1, #2 and #3) and each gradient was incubated in triplicate.

### 2.4. Measurements of Potential Denitrification, Anammox and N_2_O Production Rates

The potential nitrogen removal rates of sediments were measured using slurry incubation and isotope pairing techniques [34]. Fresh sediments were mixed with water at the ratio of 1:7 (sediment: water), and the resulting slurries were flushed with ultrahigh purity He for 30 min until an anaerobic state was reached. To remove existing NO_x_^−^ (NO_3_^−^ and NO_2_^−^) and DO, the slurries were pre-incubated in the dark at an in situ temperature (28 ℃) for 36–48 h. After pre-incubation, the slurries were transferred to 12.5 mL tubes (Exetainers, Labco, UK). The tubes were injected with the designed substrates and incubated in the incubator at in situ temperature. The final concentration of NO_3_^−^ was fixed at 100 μmol L^−^ for treatments concerning addition of DO, organic C and ammonium. The slurries incubation was thereafter terminated by adding 200 μL of 50% ZnCl2 at 0 and 2 h from the beginning of incubation. ^29^N_2_ and ^30^N_2_ signals in the tubes were detected with a membrane inlet mass spectrometry (MIMS, HPR40, Hiden, Warrington, UK). Detailed methods are described in Cai, et al. [35].

N_2_O production rates were measured with headspace equilibrium gas chromatography using the samples prepared as described above [36]. The tubes were injected with 5 mL of ultrahigh-purity He gas to replace the water phase and create headspace after inactivation and settling. Then, the tubes were violently shaken for 1 h to achieve gas-liquid equilibrium. The concentration of N_2_O in the headspace gas was detected by means of gas chromatography (GC-2014C, Shimadzu, Kyoto, Japan). Detailed methods are described in Cai, et al. [35].

### 2.5. Statistical Analysis

One-way analysis of variance (ANOVA) with Tukey’s post hoc analysis was used to test significant differences in potential rates among different concentration gradients (SPSS Statistics 24.0, IBM, Armonk, NY, USA). Linear and Michaelis-Menten kinetics were fitted to the data using the regression function of the Prism 8 software (version 8.0.2).

## 3. Results

### 3.1. Physicochemical Parameters of Water and Sediment

The overlying water of the three ponds studied had pH values ranging from 7.4 ± 1.2 to 7.6 ± 0.9 (Table 1). The concentrations of NO_3_^−^ was highly varied with values of 58.3 ± 5.8, 103.8 ± 7.3 and 10.5 ± 1.7 μmol L^−1^ in pond 1#, 2# and 3# respectively. NH_4_^+^ concentrations were determined at 7.8 ± 0.7, 24.4 ± 3.5 and 7.6 ± 1.8 μmol L^−1^ while the NO_2_^−^ concentrations in the overlying water were 3.5 ± 1.2, 7.3 ± 1.1 and 12.6 ± 4.1 μmol L^−1^, respectively.

In sediments, pH in pond 1# (6.9 ± 1.0) was lower than the two other ponds (7.5 ± 0.8 and 7.6 ±0.5). The NH_4_^+^ concentration was measured to be 13.0 ± 2.4, 21.2 ± 1.9 and 8.6 ± 1.5 mg kg^−1^ in the three ponds, and was the main form of dissolved inorganic nitrogen (DIN) accounting for 93.5, 96.4 and 93.5% of DIN, respectively. The contents of TN and TP were 2.2 ± 1.1, 3.0 ± 0.8, 6.7 ± 1.2 g kg^−1^ and 0.2 ± 0.0, 1.1 ± 0.3, 1.5 ± 0.2 g kg^−1^, in the three respective ponds. All ponds had a high OM, although the lowest value was found in pond 2# (65.2 ± 11.2 g kg^−1^), while the other two ponds were higher, i.e., 113.0 ± 15.7 g kg^−1^ in pond 1# and 105.0 ± 12.8 g kg^−1^ in pond 3#.

### 3.2. Microbial Community of Nitrogen Removal

The abundance of denitrification- and anammox-related genes in the three ponds are reported in Figure 1. The *nir* genes, genes related to denitrification, was the group of genes most highly abundant in all three ponds. *nirS* gene abundance was observed at (1.8 ± 0.0) × 10^8^, (6.4 ± 0.2) × 10^8^ and (5.2 ± 0.0) × 10^8^ copies g^−1^, for pond 1#, 2# and 3#, respectively. The abundance of *nirK* gene was determined to be (1.5 ± 0.1) × 10^8^, (5.2 ± 0.2) × 10^7^ and (5.5 ± 0.3) × 10^7^ copies g^−1^ in the three ponds, respectively. The genes related to N_2_O-reduction were of lower abundance than *nir* gene, of which *nosZ* I was found to be present at (1.1 ± 0.0) × 10^6^, (1.2 ± 0.7) × 10^6^ and (4.2 ± 0.0) × 10^6^ copies g^−1^. The abundance of the *nosZ* II gene in pond surface sediments was determined to be (2.8 ± 0.5) × 10^7^, (1.4 ± 0.3) × 10^7^ and (12.1 ± 1.7) × 10^7^ copies g^−1^. Additionally, the abundance of the anammox bacterial 16S rRNA gene was 2–4 magnitudes lower than the denitrification functional genes, which varied from 1.5 × 10^4^ to 2.8 × 10^4^ copies g^−1^.

For denitrifiers, the 40 most abundant OTUs, containing 92.4% of nirS sequences, all affiliated to Proteobacteria. At the genus level, three genera including Steroidobacter, Azoarcus and Dechloromonas were identified across the pond sediments (Figure 2a). Azoarcus was the most dominant genus in all three pond sediments (62.7%, 62.5% and 62.1% in the three respective ponds) (Figure 2c). The relative abundance of Steroidobacter ranged from 8.3% to 35.7%, while Dechloromonas constituted 29.0%, 12.4% and 2.1% of the total sequences in the three ponds, respectively. For anammox bacteria in pond sediments, the dominant OTUs (40 OTUs, covering 99.3% of the sequences) were affiliated to Planctomycetes. 16 OTUs belonged to the anammox genus Ca. Brocadia, 7 OTUs to Ca. Kuenenia, and the remaining were divided into an unclassified cluster (Figure 2b). Pond sediments were dominated by Ca. Kuenenia (91.3%, 63.7% and 73.0%, respectively). The relative abundances of Ca. Brocadia were lower than that of Ca. Kuenenia at 5.5%, 12.9% and 4.4%, respectively (Figure 2d).

### 3.3. Effect of NO_3_^−^ and NH_4_^+^ Addition

The potential rates of denitrification significantly increased from 12.4 ± 2.2 to 26.6 ± 5.9 nmol N g^−1^ h^−1^ with the addition of NO_3_^−^ from 50 to 200 μmol L^−1^ (Tukey’s, *p* = 0.011, Figure 3a). However, there was no further significant increase in denitrification rate as NO_3_^−^ concentration increased from 200 to 800 μmol L^−1^ (ANOVA, *p* = 0.463). The response of potential denitrification rates to increasing NO_3_^−^ concentrations was in accordance with the Michaelis-Menten kinetics, demonstrating a maximum denitrification rate (*V*_max_) of 39.0 nmol N g^−1^ h^−1^ and an affinity constant (*K*_m_) of 109.5 μmol L^−1^. Ponds with higher NO_3_^−^ concentrations in the overlying water had a greater response to NO_3_^−^ addition in sediment (Appendix A). In this way, pond 1# and 2#, which both had higher NO_3_^−^ concentrations in the overlying water than pond 3#, showed more significant enhancements of potential denitrification rates at a maximum 1.9 and 2.5-fold increase with NO_3_^−^ addition. In pond 3#, the maximum increase of potential denitrification rate was only 1.0-fold as NO_3_^−^ was added to the sediment.

The potential anammox rates showed no response to the addition of NO_3_^−^ (ANOVA, *p* = 0.463), and remained low at 0.3 ± 0.4 nmol N g^−1^ h^−1^ (Figure 3a) The relative contribution of anammox to nitrogen removal (ra%) was also low (<10%). With the increasing NO_3_^−^ concentration, the potential rate of N_2_O production increased from 5.7 ± 2.6 to 9.0 ± 1.4 nmol N g^−1^ h^−1^. The response of N_2_O production rates to the increasing NO_3_^−^ concentrations also fitted with the Michaelis-Menten equation demonstrating an affinity constant (*K*_m_) of 41.8 μmol L^−1^ and a maximum rate (*V*_max_) of 8.8 nmol N g^−1^ h^−1^. N_2_O production rates in pond 2#, the pond with the highest NO_3_^−^ concentration in the overlying water, were enhanced 1.6-fold, while the pond with the lowest NO_3_^−^ concentration, pond 3#, showed only a 0.3-fold increase (Appendix A).

The addition of NH_4_^+^ had no significant effects on the potential denitrification rates which ranged from 16.8 to 18.9 nmol N g^−1^ h^−1^ (ANOVA, *p* = 0.978, Figure 3b). As for potential anammox rates, no clear variation was found when the NH_4_^+^ concentration increased from 20 to 120 μmol L^−1^ (ANOVA, *p* = 0.994). The relative contribution of anammox to nitrogen removal (ra%) was low, varying from 4.8 to 11.6%. Similarly, the addition of NH_4_^+^ exerted no effect on the N_2_O production rates and the relative proportions of N_2_O to N_2_ and N_2_O (ANOVA, *p* = 0.775 and 0.958, respectively).

### 3.4. Effect of DO and Organic C Introduction

Generally, denitrification in the sediment was observed to be inhibited by the elevation of oxygen (*p* = 0.011, R^2^ = 0.40, Figure 4a). Until a concentration of 2 mg L^−1^, the introduction of DO significantly decreased denitrification rate from 20.0 to 15.1 nmol N g^−1^ h^−1^ (ANOVA, *p* = 0.019), but hereafter no significant change was observed when the DO concentration was further increased (ANOVA, *p* = 0.825). There was no significant correlation between DO and anammox rates (ANOVA, *p* = 0.986), whose rates ranged from 1.1 to 1.2 nmol N g^−1^ h^−1^. Furthermore, the relative contribution of anammox to nitrogen removal was always lower than 10%. As the DO concentration increased, N_2_O production rates showed no significant change, varying from 3.4 to 5.4 nmol N g^−1^ h^−1^ (ANOVA, *p* = 0.639). An increase in DO also had no effect on the relative proportions of N_2_O (ANOVA, *p* = 0.556).

Potential denitrification rates can be enhanced by the addition of glucose (Figure 4b). Until a concentration of 200 μmol L^−1^, glucose addition resulted in significantly higher denitrification rates, which increased from 13.3 to 23.5 nmol N g^−1^ h^−1^ (ANOVA, *p* = 0.027). No significant changes were observed with further glucose addition beyond 200 μmol L^−1^ (ANOVA, *p* = 0.791). However, the response of the potential denitrification rate to the addition of glucose in the various pond sediments was related to the OM contents in sediments (Appendix A). With the addition of glucose until a concentration of 600 μmol L^−1^, denitrification rates increased more greatly in pond 2# (0.9-fold increase) which had lower OM content, than in pond 1# and 3#, which both had higher OM contents (0.5- and 0.4-fold increase, respectively). Anammox rates ranged from 1.6 to 1.8 nmol N g^−1^ h^−1^ (ANOVA, *p* = 0.950), but showed no response to the addition of glucose. In contrast to the group without glucose addition, the N_2_O production rates were significantly higher when glucose concentration was 200 μmol L^−1^ (Tukey’s, *p* = 0.045). With the addition of glucose, no significant change to the relative proportions of N_2_O was observed (ANOVA, *p* = 0.899).

## 4. Discussion

Small waterbodies frequently receive high loads of pollution from human activities and stormwater runoff due to their small area and close proximity to human settlements [37,38]. Compared to large waterbodies, small waterbodies have relatively low buffer capacity against the impact of pollution discharge and stormwater runoff, and thus their physicochemical properties may shift dramatically [13]. Among these physicochemical properties, the dissolved inorganic nitrogen (DIN), organic C and dissolved oxygen levels are particularly important for the microbial nitrogen cycle [39,40,41].

Our study showed that the NO_3_^−^ was the most important parameter altering the microbial nitrogen cycle in sediment. The artificial addition of NO_3_^−^ significantly increased the potential denitrification rates 1.8-fold when the NO_3_^−^ concentration increased from 50 to 800 μM. These results suggest that the denitrification process in such small waterbodies was limited by NO_3_^−^, which are consistent with studies conducted on rivers, estuary sediments and paddy soils as the addition of NO_3_^−^ alleviates the limitation of substrate deficiency process [21,42,43]. However, in contrast with rivers and estuaries, in natural conditions, these changes are prone to occur in small water bodies, whose environments may be dramatically influenced by runoff confluence [13]. Furthermore, the denitrification genes (*nirS*, *nirK*, *nosZ* I and *nosZ* II) were detected with high copy numbers (10^6^–10^9^ copies g^−1^) in pond sediments, being higher than those reported for marine and freshwater sediments [44,45,46,47,48,49]. Previous studies found that the higher the NO_3_^−^ concentrations the better the denitrification genes expressed [50]. Thus, for pond sediments, elevated NO_3_^−^ may induce more expression of denitrification genes, leading to more NO_3_^−^ being reduced to gas and released. Since the anammox rates remained low in the present study, we can conclude that the nitrogen removed mainly occurred through added NO_3_^−^ via denitrification. In the meantime, denitrification functional genes were 2–4 magnitudes more abundant than the anammox functional genes, which could explain the domination of denitrification process. Moreover, the sediment with a lower NO_3_^−^ concentration in the overlying water (pond 3#) showed a lower *V*_max_ for denitrification with NO_3_^−^ addition. This can be explained by microbial denitrifiers in poor NO_3_^−^ habitats being well-adapted to such environments, and so their denitrification rates respond weakly to NO_3_^−^ additions [51]. There were no significant differences in the abundance of denitrification genes among any of the three ponds. The contrasting response to NO_3_^−^ addition of the three ponds sediment may be ascribed to their community compositions which showed significant difference [52]. The addition of NH_4_^+^ exerted no effect on denitrification and anammox rates in pond sediments. This is probably due to NH_4_^+^ usually being in surplus and is therefore not a limiting factor for denitrification in sediments [53].

The addition of glucose was found to promote the denitrification rates 0.8-fold when the concentration was increased to 200 μmol L^−1^. Organic C acts as an electron donor in heterotrophic denitrification [39]. Therefore, it can be suggested that organic C plays a significant role in promoting denitrification in various habitats [21,54]. However, it should be noted that the ^15^NO_3_^−^ was also added into the simulation system (final concentration at 100 μmol L^−1^) to measure the denitrification and anammox rates in the test of organic C addition. As the substrate of denitrification, NO_3_^−^ often acts as a limiting factor in such habitats [55]. Therefore, the addition of organic C only may not promote denitrification when NO_3_^−^ is deficient in situ. On the contrary, DO elevation exerted negative effects on denitrification. Lower DO levels create ideal reduction conditions for the denitrification process [56,57]. Furthermore, NO_2_^−^ reductase (encoded by *nirS* and *nirK* gene) could be inhibited by high DO concentrations, leading to the accumulation of NO_2_^−^, which consequently limits denitrification [58,59,60]. However, the presence of oxygen may enhance the ammonia oxidation processes that provide substrates (NO_2_^−^) for denitrification and anammox. Although the nitrogen removal from the sediment was inhibited immediately after the elevation of DO, the nitrogen could still be removed in the long term.

N_2_O emission during denitrification is also of concern. The denitrification genes *nir* (N_2_O producer) and *nosZ* (N_2_O reducer) are often used as genetic markers for N_2_O production [61]. The proportion between N_2_O-producing and N_2_O-reducing microorganisms (*nir*/*nosZ* ratio) can partly explain N_2_O emissions, which were positively correlated in some studies [62,63]. Nevertheless, negative correlation between the *nir*/*nosZ* and N_2_O/(N_2_O  +  N_2_) ratio were also observed in previous studies [64,65]. Therefore, N_2_O emissions are not only controlled by genetic potential, but also by transcriptional regulation and enzymatic activity [66]. However, environmental factors play an essential role in the processes. In this study, NO_3_^−^ also significantly influenced the N_2_O production not only denitrification. Although N_2_O production increased, the proportion of N_2_O production in nitrogen loss (N_2_O/(N_2_ + N_2_O)) decreased. The addition of NO_3_^−^ may promote the expression of N_2_O production and reduction genes [50], so N_2_O increased less than N_2_. In other words, the addition of NO_3_^−^ not only promoted nitrogen removal, but also reduced greenhouse gas emissions, which is more environmentally beneficial. It was found that the addition of C sources such as glucose increased the abundance and expression of *nosZ* gene, but had no effect on *nirS* gene [67,68]. Furthermore, the introduce of DO could inhibit NO_2_^−^ reductase, which consequently reduced production of N_2_O and N_2_ [58,59,60].

External inputs usually carry NH_4_^+^, DO, NO_3_^−^ and organic C from domestic pollution, rainfall runoff, and agricultural pollution to waterbodies. The present results showed that NO_3_^−^ and organic C promoted denitrification and nitrogen removal from water sediments until a certain extend. Although DO inhibited denitrification in the short term, nitrification could be activated over time, which supplies substrates for the denitrification process and can thereby promote nitrogen removal. Since NH_4_^+^ in the sediment was in surplus, the addition of NH_4_^+^ showed no significant effect on nitrogen removal from the pond. Small waterbodies play an essential role in agricultural watersheds to store nitrogen and mitigate the pollution output from the watersheds [35]. NO_3_^−^ is the dominant form of nitrogen in runoff [69], primarily due to mineralization and nitrification of nitrogen fertilizers in agricultural watersheds [70], and the higher mobility of NO_3_^−^ [7]. Our study demonstrated that rainfall runoff into the pond may create a “hotspot moment” of microbial nitrogen removal, thereby reducing nitrogen pollution in the watershed.

## Figures and Tables

**Figure 1 microorganisms-10-01429-f001:**
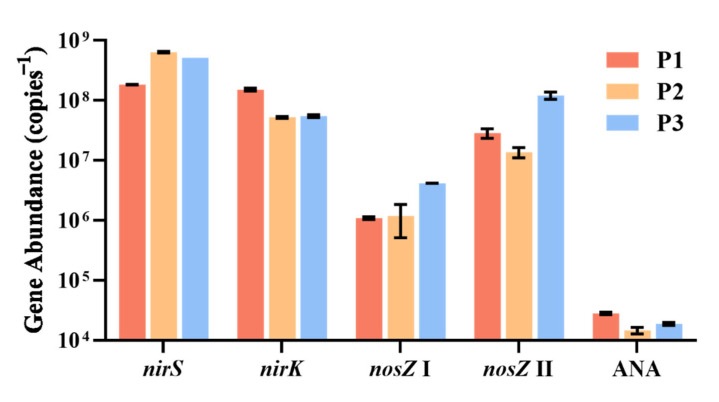
Abundances of nitrogen removal related genes including *nirS*, *nirK*, *nosZ* I, *nosZ* II and anammox 16S rRNA in sediments of the three ponds (Pond 1#–Pond 3#).

**Figure 2 microorganisms-10-01429-f002:**
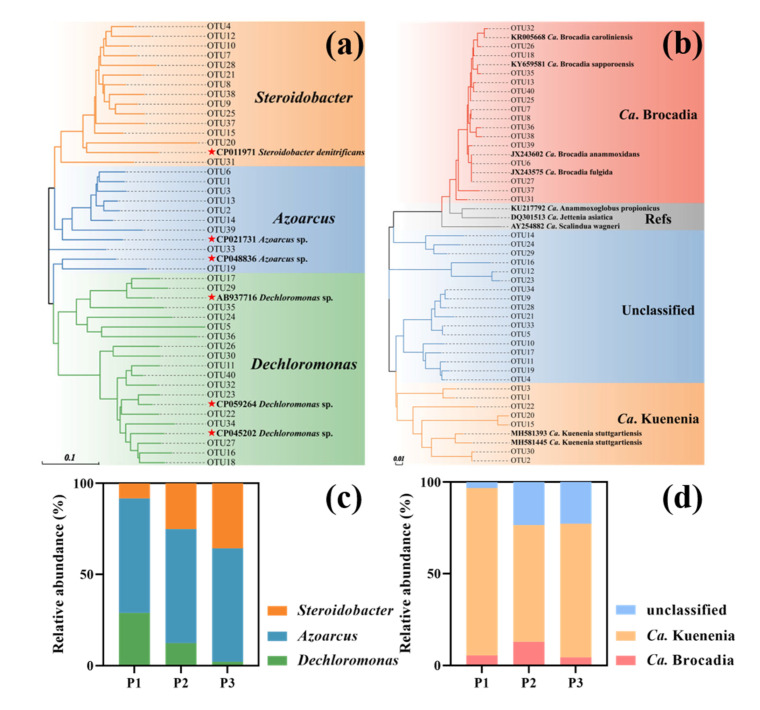
Neighbor-joining phylogenetic tree of the most abundant (**a**) *nirS* and (**b**) Anammox-specific 16S rRNA gene OTUs in the three pond sediments (P1–P3) and the reference sequences from GenBank. Bootstrap values were 1000 replicates. Relative abundance of (**c**) *nirS*-type denitrifiers and (**d**) anammox at the genus level.

**Figure 3 microorganisms-10-01429-f003:**
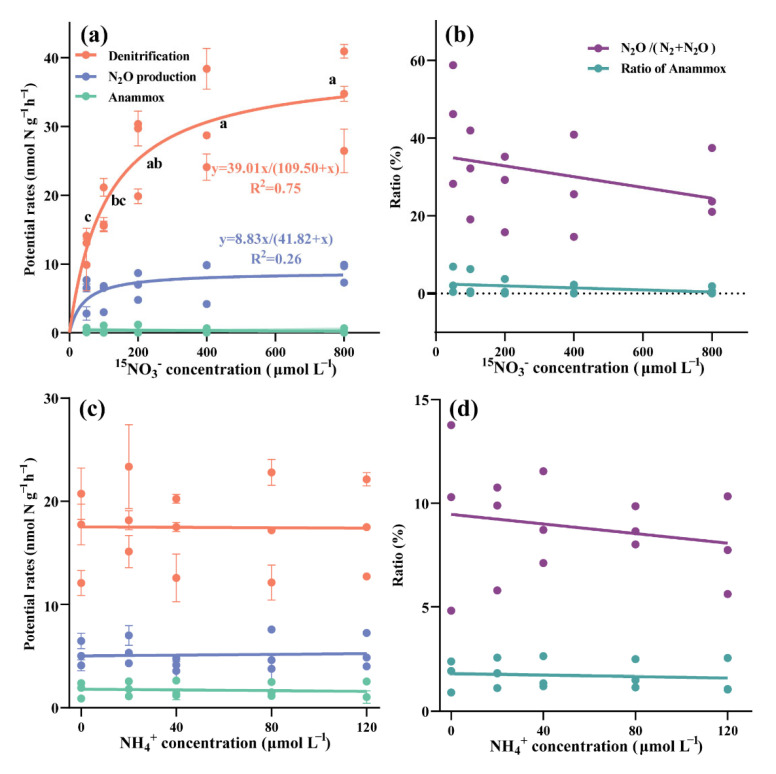
Potential rates of denitrification, N_2_O production and anammox during denitrification under varying concentrations of (**a**) NO_3_^−^ and (**c**) NH_4_^+^, and relative contribution of anammox to nitrogen removal and relative proportion of N_2_O under varying concentrations of (**b**) NO_3_^−^ and (**d**) NH_4_^+^. Different letters indicate significant differences among treatments (Tukey test, *p* < 0.05).

**Figure 4 microorganisms-10-01429-f004:**
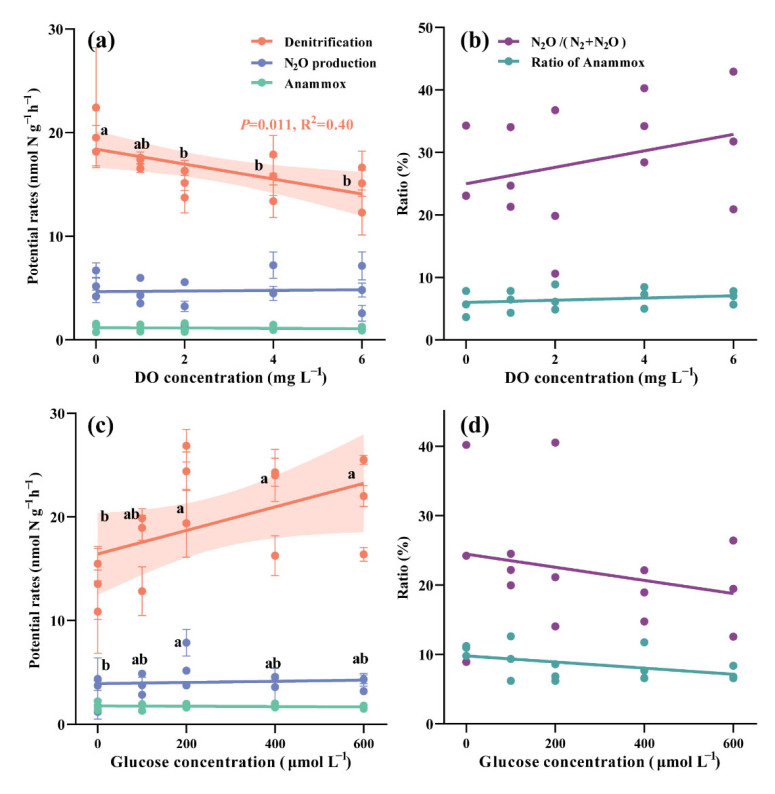
Potential rates of denitrification, N_2_O production and anammox during denitrification under varying concentrations of (**a**) dissolved oxygen (DO) and (**c**) glucose, and relative contribution of anammox to nitrogen removal and relative proportion of N_2_O under varying concentrations of (**b**) dissolved oxygen (DO) and (**d**) glucose. Values with different letters indicate significant differences among the treatments (Tukey test, *p* < 0.05).

**Table 1 microorganisms-10-01429-t001:** Physicochemical characteristics of three ponds.

	Site	Pond 1#	Pond 2#	Pond 3#
**Overlying water**	pH	7.4 ± 1.2	7.6 ± 0.9	7.5 ± 0.1
	NH_4_^+^ (μmol L^−1^)	7.8 ± 0.7	24.4 ± 3.5	7.6 ± 1.8
	NO_3_^−^ (μmol L^−1^)	58.3 ± 5.8	103.8 ± 7.3	10.5 ± 1.7
	NO_2_^−^ (μmol L^−1^)	3.5 ± 1.2	7.3 ± 1.1	12.6 ± 4.1
**Sediment**	Moisture (%)	73.8 ± 6.3	45.0 ± 5.0	59.4 ± 6.1
	pH	6.9 ± 1.0	7.5 ± 0.8	7.6 ± 0.5
	NH_4_^+^ (mg kg^−1^)	13.0 ± 2.4	21.2 ± 1.9	8.6 ± 1.5
	NO_3_^−^ (mg kg^−1^)	0.6 ± 0.0	0.5 ± 0.1	0.4 ± 0.1
	NO_2_^−^ (mg kg^−1^)	0.3 ± 0.1	0.3 ± 0.0	0.2 ± 0.0
	DIN (mg kg^−1^)	13.9 ± 2.5	21.9 ± 1.9	9.1 ± 1.8
	TN (g kg^−1^)	2.2 ± 1.1	3.0 ± 0.8	6.7 ± 1.2
	TP (g kg^−1^)	0.2 ± 0.0	1.1 ± 0.3	1.5 ± 0.2
	OM (g kg^−1^)	113.0 ± 15.7	65.2 ± 11.2	105.0 ± 12.8

## Data Availability

All supporting data was submitted as Appendix A.

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
