# Peer review of "Nitrate Addition Increases the Activity of Microbial Nitrogen Removal in Freshwater Sediment"

_microorganisms, 2022, doi:10.3390/microorganisms10071429_

Round 1

Reviewer 1 Report

This paper reports the removal of added nitrate with various concentrations in freshwater sediment when mixed with water. This kind of information is important for considering the local changes in the nitrogen cycle in natural environments. However, the presented experiments were performed under conditions far from the natural sediments, since the collected sediments were mixed with water at the ratio of 1:7 (sediment: water), and the environment of microbes in the mixture should be largely different from original environments. In addition, the degree of mixing is ambiguous, and sediments are thought to be dispersed in uncontrolled various sizes of particles. Without a detailed examination of the state of the mixture and the difference from the original sediments, the meanings of the obtained values are hard be explained. In this point of view, the presented experimental data are too premature to be published in "Microorganisms. Additional two major comments are as follows. (1) The authors mentioned that the results of the presented experiments in small waterbodies are consistent with reported studies on rivers, estuary sediments and paddy soils. However, the similarity and difference are not adequately discussed. If just consistent, the presented amount of data is not worth for an additional publication.The previous publications should  also be mentioned in the introduction. (2) The authors presented data on the abundances of genes for denitrification, but few significant new findings are apparent in the data, and little new discussion is presented.

Author Response

Thank you very much for handling our manuscript.

We took a lot effort revising the manuscript and responding to all the concerns in the past few days. Great amendment was made, especially in the section of M&M and Discussion. Please find our responses in the attachment and hope the revised MS could meet the criteria of Microorganisms.

Best regards,

Yu Wang and co-authors

Reviewer 2 Report

Current work states pollutants such as nitrate removal application through anammox and deni affecting COD and nitrogen loss and removal. Experiments showed a significant removal effect by nitrate and bacterial cultivation significantly improved with Nitrate addition. – Give standard deviation of your results, mean, max and min. Give measure of ammonia and nitrite, too

What about nitrogen losses and increases in N2O emissions, were emissions measured and theoretically estimated additionally? This paper presents the latest developments on the natural water body N removal system based on energy efficient processes using new insights into N biotransformation involving metabolic pathways, its elimination and functional microorganisms involved during the formation of process.

„Physicochemical properties of NH4 + , NO2 - , NO3 - and pH in overlying water and NH4 + , NO2 - , NO3 - , TN (total nitrogen), TP (total phosphorus), TOM (total organic matter), pH  and Moisture content in sediments were determined and described in Cai, et al. [26]“- these are standard methods needing ref to standard methods

Term „TOM (total organic matter)“ is not often used, try DOM or TOC

Main aims of studies need to be stated point by point along with hypothesis

„The introduction of DO significantly inhibited denitrification until a concentration of 2 mg L-1 , after which no additional inhibition was observed“- Check that if your statement has not been wrong compared to other studies DO affects from low contents limiting denitrification

Figures quality is bad and they could improve their amount. Influent and effluent nitrogen forms need to be added simultaneously to tests result graphs. Sharpen graphs up, too and add larger fonts. Error bars on Fig. 1 seem to be halfly missing, to see any significant difference add correctly to everywere.

Caption and a lot of writings not needed within Figures

Legend need to be inside figure area.

Check caption and spacing mistakes otherwise hard to understand? Such as, in Figure y axis, spaces  needed after parameter , before units

P values and standard deviations should be included in Your MS stating before the normality test was it done beforehand. ANOVA and tukey test should not be mixed.

Concrete microbial strains need to be added to MS with bacterial specimens names, too-

After „ Although several studies have correlated environmental factors to microbial nitrogen removal processes in natural habitats, the means by which these factors regulate the intensity of denitrification and anammox activities in small waterbodies remain unknown “ some works can be cited: Literature has shown different environmental MBBR and other treatments to be solved for more economical way, which could be shown: DOI: 10.1016/j.biortech.2016.02.051, https://doi.org/10.1016/j.scitotenv.2021.149133, https://doi.org/10.3390/w13030350, https://doi.org/10.1080/09593330.2013.874492, https://doi.org/10.1016/S1001-0742(10)60523-2, https://doi.org/10.1080/09593330.2020.1721566, https://doi.org/10.1080/09593330.2014.941946

Three main N2O production processes, namely hydroxylamine oxidation, nitrifier denitrification, and heterotrophic denitrification, and the unique  N2O consumption process, namely nosZ-dominated N2O degradation, in the anammox-driven systems need to be summarized and discussed. The key factors influencing N2O  emission and mitigation strategies were not discussed in detail, and areas in which further research urgently required are needed to be identified.

The treatment performance, enzymes NosZ and microbial communities functional effects should be studied, nitrogen removal mechanism to be analyzed.

Author Response

(The authors gave the same response as above.)

Round 2

Reviewer 1 Report

The authors have considered the reviewer's comments seriously and revised the manuscript accordingly as far as they can do at present. As for the first point concerning the appropriateness of the research strategy, the reviewer still does not agree with the authors' opinions, but significant potential readers may have positive opinions on the authors. Therefore, I agree to publish the paper in Microorganisms in the present form.